# What Is the Exact Contribution of *PITX1* and *TBX4* Genes in Clubfoot Development? An Italian Study

**DOI:** 10.3390/genes13111958

**Published:** 2022-10-27

**Authors:** Anna Monica Bianco, Giulia Ragusa, Valentina Di Carlo, Flavio Faletra, Mariateresa Di Stazio, Costantina Racano, Giovanni Trisolino, Stefania Cappellani, Maurizio De Pellegrin, Ignazio d’Addetta, Giuseppe Carluccio, Sergio Monforte, Antonio Andreacchio, Daniela Dibello, Adamo P. d’Adamo

**Affiliations:** 1Genetics Unit, Institute for Maternal and Child Health, IRCCS “Burlo Garofolo”, 34137 Trieste, Italy; 2Unit of Paediatric Orthopaedic and Traumatology, Institute for Maternal and Child Health, IRCCS “Burlo Garofolo”, 34147 Trieste, Italy; 3Unit of Pediatric Orthopaedics and Traumatology, Istituto Ortopedico Rizzoli (IRCCS), 40136 Bologna, Italy; 4Pediatric Orthopedic San Raffaele Hospital of Milano, 20132 Milan, Italy; 5Unit of Pediatric Orthopaedics and Traumatology Giovanni XXIII Children’s Hospital, Via Giovanni Amendola, 70126 Bari, Italy; 6Pediatric Orthopedic of Buzzi Children Hospital of Milano, 20154 Milan, Italy; 7Department of Medical, Surgical and Health Sciences, University of Trieste, 34100 Trieste, Italy

**Keywords:** clubfoot, congenital talipes equinovarus, congenital malformation, *TBX4*, *PITX1*

## Abstract

Congenital clubfoot is a common pediatric malformation that affects approximately 0.1% of all births. 80% of the cases appear isolated, while 20% can be secondary or associated with complex syndromes. To date, two genes that appear to play an important role are *PTIX1* and *TBX4*, but their actual impact is still unclear. Our study aimed to evaluate the prevalence of pathogenic variants in *PITX1* and *TBX4* in Italian patients with idiopathic clubfoot. *PITX1* and *TBX4* genes were analyzed by sequence and SNP array in 162 patients. We detected only four nucleotide variants in *TBX4*, predicted to be benign or likely benign. CNV analysis did not reveal duplications or deletions involving both genes and intragenic structural variants. Our data proved that the idiopathic form of congenital clubfoot was rarely associated with mutations and CNVs on *PITX1* and *TBX4*. Although in some patients, the disease was caused by mutations in both genes; they were responsible for only a tiny minority of cases, at least in the Italian population. It was not excluded that other genes belonging to the same *TBX4-PITX1* axis were involved, even if genetic complexity at the origin of clubfoot required the involvement of other factors.

## 1. Introduction

One of the most common pediatric congenital limb deformities is the Congenital Talipes Equinovarus (CTEV), often known as clubfoot, affecting an estimated 1/1000 live births. However, this rate varies among different countries around the world [1]. CTEV is a malformation characterized by a torsion of the longitudinal axis of the foot, a consequence of a malalignment of the talocalcaneonavicular complex, and is responsible for significant disabilities in children.

There are three forms of clubfoot: mild forms with no need for treatment, intermediate forms that require standard cures, and complex forms with additional neurological malformations, which need additional treatments.

In clubfoot, there are all 4 components, expressed in different degrees of severity:Metatarsus adductus: the fingers point inside with concavity of the medial foot margin;Hindfoot varus deformity: medial deviation of the longitudinal axis of the calcaneus;Midfoot cavus deformity: the sole is rotated upwards;Hindfoot equinus deformity: extreme and irreducible plantar flexion.

There are three clinical types of congenital clubfoot: talipes equinovarus, talipes calcaneovalgus and metatarsus varus. Within them are several subgroups with different clinical associations and aetiology [2,3,4,5,6,7]. The condition can be bilateral in about half of the patients and occurs twice as frequently in infant boys than in girls [8]. In unilateral cases, the compromised foot is often the right one, and the affected foot is shorter with less calf muscle bulk than usual [8,9,10,11]. Several classification systems have been proposed for determining the severity of the foot defect, such as the Manes, Costa classification, the Dimeglio score [12] and the Pirani score [12]. Manes-Costa classification is a system based on three severity degrees (grade I: the deformity is mild and ultimately reducible, grade II: the deformity is moderate and only partially reducible and grade III: the deformity is severe and not reducible) that classify the severity of the CVTE deformity exclusively on the sagittal plane [13].

On Dimeglio score, we considered four grades of clubfeet: grade I, called “soft-soft feet” (benign feet, similar to postural feet), with a score of 5 to 1, grade II, named “soft > stiff feet” with a score 5–10 (moderate feet, reducible but partly resistant), grade III nominated “stiff > soft feet” with score 10–15 (severe feet, resistant but partly reducible) and grade IV designated “stiff-stiff feet” with score 15–20 (very severe, pseudo arthrogryposis feet). The scale of 0 to 20 was established based on four essential parameters: equinus in the sagittal plane, varus deviation in the frontal plane and derotation around the talus of the calcaneo-forefoot block and adduction of the forefoot on hindfoot in the horizontal plane [14].

The Pirane score is a simple system to determine the severity and treatment progress in children with CVTE. It works on clinical signs of contracture and consists of 6 parameters (3 in each in midfoot and hindfoot) which are scored as no deformity (0), moderate (0.5), severe deformity (1 to 6) [12,15].

When its presence is associated with other congenital anomalies (20% of patients) can be syndromic or secondary. On 80% of patients occur without other malformations as an isolated congenital disability, and it is classified as Idiopathic Congenital Talipes Equinovarus (CTEV) [16]. The aetiology is yet unknown, and several causative factors can be involved, such as nerve lesion, muscular abnormality, vascular defect, neuromuscular defect, regional growth disturbance, intrauterine extrinsic pressure induction and genetic component alone or together with environmental interaction [2,3,4,5,6]. Genetic malformations of the limbs are characterized by genetic heterogeneity and several different phenotypic features. Congenital clubfoot aetiology could be considered a mix of environmental and genetic factors with a higher recurrence within families. Engell et al. suggested, in a twin study, that the genetic factor might contribute, but the environmental factor must play an essential role in the aetiology of the congenital clubfoot [17].

### PITX1-TBX4 Axis

Despite advances in clubfoot genetics, few genes are known to be involved in determining the phenotype. To date, the *PITX1*-*TBX4* pathway, in which proper function is necessary for normal hindlimb development, is described in the literature as the most strongly associated with the malformation [18]. Mutations in both genes have been described. The isolated clubfoot results in causative heterozygous variations, hence dominant mutations in the gene encoding *PITX1*. The clubfoot phenotype is also described in patients with *TBX4* microdeletions. Moreover, copy number variations in both genes have been associated with the clubfoot phenotype [19,20,21,22].

Several studies support the evidence of the crucial role of *PITX1* (MIM 602149) and *TBX4* (MIM 601719) genes in early limb development and clubfoot aetiology. *PITX1* and *TBX4* encode two transcription factors expressed overall in the developing hindlimb: *PITX1* is involved in hindlimb identity [18], and *TBX4* is a hindlimb-specific T-box transcription factor participating in the patterning of limb’s muscles and tendons [23].

In 2008, a single missense variant with AD inheritance and reduced penetrance (c.G388A; p.E130K) was identified in the *PITX1* gene in all affected members of a North American family with clubfoot [19]. The affected individuals showed unilateral or bilateral clubfoot with additional phenotypes, such as bilateral preaxial polydactyly, development of hip dysplasia or patella’s bilateral hypoplasia. Incomplete penetrance was observed in five carrier females of the family. Functional studies performed on the missense genetic variant, located on a highly conserved homeodomain, showed a reduced ability to transactivate a luciferase reporter, resulting in a dominant negative effect on transcription [19].

In another patient with CTEV, right tibia hemimelia, and bilateral preaxial polydactyly, a frameshift mutation was identified due to a heterozygous deletion c.765_799del35; p.A256Rfs on *PITX1* gene [24]. Moreover, by copy number variation analysis in a patient with isolated familial clubfoot, a microdeletion of 241bp involving the *PITX1* gene was identified. The deletion was present in over three generations as an autosomal dominant disorder. The first genetic *pitx1* mouse model allowed defining a *PITX1* haploinsufficiency as a genetic mechanism for clubfoot [25].

Regarding the *TBX4* gene, it is a member of the T-box transcription family that plays a significant role during embryogenesis [26] and its expression is regulated by the *PITX1* targets enhancer HLEA or HLEB. In *TBX4*, the gene copy number variations (CNV) and heterozygous single nucleotide variants have been associated with several phenotypes, such as Acinar dysplasia and ischiocoxopodopatellar syndrome with or without pulmonary arterial hypertension. In a family with clubfoot associated with additional developmental anomalies, a 2.15 Mb duplication on chr17 involving the *TBX4* gene was detected; while in another family with isolated CTEV, a microduplication of 350kb was identified on the chr17_q23.123.2 encompassing the *TBX4* gene and its enhancers. Three CTEV unilateral or bilateral families with mild skeletal abnormalities and short stature inherited an autosomal dominant trait with incomplete penetrance were found in heterozygous a 2.2 Mb microduplication in 17q23.1q23.2 involving *TBX4* [20,21,27,28].

In this study, to evaluate the real impact of mutations in these 2 genes in the aetiology of ICTEV, we analyzed a total of 162 severely isolated idiopathic clubfoot patients with at least one affected first-degree relative. Furthermore, given the extreme heterogeneity underlying ICTEV, we selected patients from different Italian centers to avoid bias on allelic frequencies due to the origin population. On 17 patients, we could also analyze some family members to identify potential family forms. *PITX1* and *TBX4* genes were analyzed by Sanger sequencing and SNP-array for all patients and family members.

## 2. Materials and Methods

### 2.1. Ethics Statement

All patients with clubfoot were recruited at the Institute for Maternal and Child Health- IRCCS “Burlo Garofolo” of Trieste. The institutional Ethical Committee approved this study, and all patients, parents or legal representatives provided informed consent. The blood sample collection was consistent with the guidelines of experimental research.

### 2.2. Patients

Patients were diagnosed with talipes equinovarus based on the physical examination. To avoid bias related to the frequency of alleles in specific populations, we conducted a monocentric study by collecting patients from different Italian regions: 88 Patients from the Department of Pediatric Orthopedic and Genetic of “IRCCS” Burlo Garofolo of Trieste, 49 patients from the Department of Pediatric Orthopedic of Rizzoli of Bologna, 5 patients from the UOS of pediatric orthopedic San Raffaele Hospital of Milano, 17 patients from the Department of Pediatric Orthopedic of Giovanni XXIII Children Hospital of Bari and 3 patients from the Buzzi Children Hospital of Milano.

Some of these centers are of reference for treating orthopedic diseases in Italy, also admitting patients from other regions. The subjects were enrolled from 2016 to 2021 according to the following inclusion criteria: isolated clubfoot talipes equinovarus, idiopathic clubfoot, monolateral and bilateral clubfoot, but also patients with only severe form of clubfoot such as Manes-Costa classification 3, Pirani score 5 or 6 and children who underwent to an Achilles tendon tenotomy for the severity of their clubfoot. Appendix A show families (proband with parents) patients, while Appendix A shows only the individual patients.

The exclusion criteria were clubfoot other than CTEV, subjects with additional congenital anomalies and/or developmental delay.

### 2.3. Samples Collection

The blood samples were collected from affected individuals and affected or unaffected family members when available. The phenotypic data were tracked and collected for each patient, parent and family member.

### 2.4. DNA Extraction

The genomic DNA of each patient was extracted from peripheral blood by QIAsymphony workstation and QIAsymphony DNA kits (Qiagen, Milan, Italy). For all DNA samples collected, the concentration was measured with Nanodrop 2000 spectrophotometer (Thermo Fisher Scientific, Waltham, MA, USA). The standard DNA samples were stored in refrigeration at −20 °C.

### 2.5. Primer Design

To analyze the coding sequence of *PITX1* and *TBX4* genes, the online tool NCBI National Centre for Biotechnology Information (NCBI) Primer Blast was used for primer designing (Appendix A). Primer pairs were designed in the intron sequences flanking for each one of the exons to screen coding sequences and splicing signals (primer sequences in Appendix A). Amplimers, amplified by KAPA2G FAST HS RM (Sigma-Aldrich), were checked for quality in agarose gel and sent, after purification, for bidirectional DNA sequencing service to Macrogen Europe in Amsterdam. Sequences were retrieved, aligned, analyzed, and reviewed using Codon Code Aligner 7.1.1 and 4Peaks programs.

### 2.6. CNV Analysis

Genotyping was performed by Illumina Infinium Global Screening Array v.3.0 using the Human OmniExpressExome-8 Chip containing 960.919 loci (Illumina, San Diego, CA, USA). Log Ratio (LRR). Frequency values of B allele were (BAFs), and log R were generated by Illumina’s GenomeStudio v2.0.1 and CNVpartition software (Illumina, San Diego, CA, USA) on the GRCh37/hg19 Human build. The copy number variants, mapped to the GRch37/hg19, were annotated with UCSC RefGene and data was exported from GenomeStudio to a Postgres Relational Database.

## 3. Results

One hundred and sixty-two patients were included: 123 male (76%) and 39 female (24%). Among them, there was a group with only clubfoot talipes equines varus, with idiopathic clubfoot, 1 monolateral, 60 either bilateral, and other patients presented a more severe form of clubfoot such as Manness-costa classification 3, Pirani scores 5 or 6 and finally children that underwent to an Achilles tendon tenotomy for the severity or their clubfoot.

### 3.1. Result of PITX1 and TBX4 Gene Sequencing

To identify the *PITX1* and *TBX4* mutations responsible for clubfoot, we sequenced all the coding regions and exon-intron boundaries of 162 patients’ samples. No evident mutations were detected on both gene, and no variant has been identified in the *PITX1* genes.

#### *TBX4* 

We found at least one variant in 56 patients (34.7%); in 23 patients, a combination of 2 variants and in 8 patients (5%), 3 variants (Appendix A).

All the identified variants were predicted as benign and not associated with clubfoot onset predisposition in literature.

The variants identified were:rs3744448 (MAF:0.1675; c.G17C;p.G6A), detected in 35 our cases. In our patients, this variant was found to be a single polymorphism and associated with two other SNPs [29,30].The common variant rs3744447 (c.T276G; p.A92A, MAF in gnomAD 0.18) was identified in 31 patients.rs758596, an intronic variant (c.402-8G>A; MAF in gnomAD 0.31) was present in 60 patients.rs777880490 (c.C24T; p.S8S), a synonymous variant with an interesting, very-low-frequency in the general population (MAF: 0.00008714) and saw only in one our patient together with the polymorphism rs758596.

Interestingly, all these variants were tested at Illumina Laboratory Services as part of a predisposition screening for Coxopodopatellar syndrome in a healthy population.

An automated score was calculated for all these cases using the variant allele frequency, disease prevalence, and estimates of penetrance and mode of inheritance to assess whether this variant is too frequent to cause disease. Based on the score and the internal cut-off, a variant classified as benign is not subjected to further treatment. The score for this variant led to a classification of benign for this disease. We take data about these polymorphisms on the following sites https://clinvarminer.genetics.utah.edu (accessed on 28 June 2022) and https://gnomad.broadinstitute.org/ (accessed on 28 June 2022).

### 3.2. CNV Analysis

To understand if genomic copy number variations (CNV) could be responsible for clubfoot probands, they were analyzed for CNVs with Illumina SNP array. We did not find any CNV on both *TBX4* and *PITX1* genomic regions.

## 4. Discussion

Several cases of clubfoot are not associated with other structural abnormalities. It is a severe congenital disability that affects approximately 0.1% of all infants. There is strong evidence for genotype-phenotype correlation for isolated clubfoot, but much of the hereditability remains unexplained. The extracellular matrix, connective tissue components, the contractile muscular apparatus, and cytoskeletal genes are associated and linked with clubfoot aetiology.

Several genetic studies consider genes involved in the *PITX1-TBX4* regulatory axis as the most significant candidate genes [22]. On *TBX4,* only deletions or duplications were found in patients with CTEV. Duplications involving *TBX4* [28] and its regulatory elements are considered strong candidates for ICTEV. So, no point mutations but only deletions and/or duplications were found in *TBX4* on CTEV families. However, as the variable penetrance has not yet been explained, probably other factors could be involved in the CTEV aetiology [31].

In *PITX1* gene in all affected members of a North American family with clubfoot, a single missense variant with AD inheritance and reduced penetrance was identified [19].

In another patient with CTEV, a frameshift mutation was identified [24]. In a patient with isolated familial clubfoot, a microdeletion present in over three generations involving the *PITX1* gene was identified [25].

Despite these sporadic findings, the true impact of variants in these 2 genes in clubfoot determination is not yet known.

From coding gene regions sequencing and CNV-specific analysis on the genomic regions of the *PITX1* and *TBX4* genes, respectively, in the cohort of our ICTEV patients, neither a pathogenetic gene mutation nor a pathogenetic CNV was detected. Although we did not observe any mutations in the coding regions in the study, the question remains as to what the role of both *TBX4* and *PITX1* genes may be in ICTEV. There is substantial evidence in the aetiology of clubfoot of different pathways and/or families of genes, such as the *PITX1-TBX4* axis, the *HOX* family, the muscle contractile protein, and the caspases. However, the leading candidate has not yet been identified.

In the presence of limited literature on the ICTEV mutation spectrum, the present study reports the results of the complete coding region sequence analysis and the CNV analysis of the *TBX4* and *PITX1* genes. Only non-significative polymorphisms were detected in our patients, so they did not have the mutations identified in these genes in patients described in the literature suggesting the role of other genetic variants involved in idiopathic clubfoot onset.

In our cases, as well as in other patients with a clinical diagnosis of ICTEV but without the identification of the genetic cause, it remains a challenge not only to understand the exact genetic aetiology useful in determining the best therapeutic strategies but also why the identification of the gene or genes involved could help prognosis for other children if present in the family.

## 5. Conclusions

Although several studies have tried to identify the molecular basis of clubfoot and have uncovered some remarkable variations in candidate genes, the “root cause” of the disease remains unknown. The high heterogeneity between families makes it difficult to identify a common root cause, and the regulatory and non-coding part of the genome plays a vital role.

The evidence supporting a dysregulation of the *PITX1-TBX4* axis seems quite convincing. However, as demonstrated by this single-center study, only a few patients with idiopathic clubfoot have mutations in these 2 genes.

Considering the minimal impact of the 2 genes analyzed, the next step to look for additional genetic loci related to the onset of Congenital Talipes Equinovarus (CTEV) is to recruit more patients with a homogeneous phenotype to perform a whole genome association analysis. Consequently, it is essential to continue to recruit patients and extend collaboration to other clinical centers.

Future studies will have to focus on the entire pathway verifying the presence of more hypomorphic variants in some patients, analyzing genomic domains that regulate a plurality of genes such as TADs (Topologically Associated Domains) and using animal models. Our knowledge of genome regulation is still quite incomplete, making the analysis more problematic but probably the best approach to understanding the aetiology of a complex pathology like this. Whole Genome Sequencing (WGS) technology has already proved very useful in many complex pathologies. It will undoubtedly be applied in the future, also in the context of clubfoot, to continue these investigations and analyze variants outside the coding region of the genes and structural anomalies.

## Data Availability

The study’s data are available upon request from the corresponding author.

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
