# Peer review of "What Is the Exact Contribution of PITX1 and TBX4 Genes in Clubfoot Development? An Italian Study"

_genes, 2022, doi:10.3390/genes13111958_

Round 1

Reviewer 1 Report (New Reviewer)

Since it is established in literature that the disruption of TADs and creation of neo-TADs are dominant mechanisms of SVs (e.i. PMID: 25959774; PMID: 27706140) and there are some studies suggesting the involvement of TADs in the limbs development (e.i. PMID: 33497014, PMID: 31511252, PMID: 30475793), it would be interesting to developed a little more the importance of the disruption of TAD as cause for undiagnosed genomic diseases, specially CTEV.

Since you have a well-characterized group of patients, there are several genes already associated to limbs development, and WGS is currently a reliable and relatively robust methodology to be applied in clinical diagnosis, after the laboratory goes through a stage of validation and confirmation of pathogenic variants by other methodologies, it would be interesting to apply it to this group.

All the text should be revised, since it were detected some inaccuracies:

line 55, 83, 96, 151 - small mistakes

line 149 to 155 – harmonize the way to designate the departments (lower/capital letters)

line 163 and 164 - harmonize the way how to refer the tables (Table number or just Table) 

line 179 - the table 1 refereed corresponds to “Variants identified in TBX4 gene. All are predicted as benign or likely benign. Three in coding region (Two synonymous and one missense) and one intronic. MAF: Minor Allelic Frequency” not making sense in the text.

line 265 to 266 - the sentence doesn't make sense

Supplementary material

The "Table S2 - Family’s cases" is confused and difficult to consult. Should be improved.

The "Table S3 - Isolated probands" should have a logical order for easy consultation (e.i., by order of "cod_pt", for example).

Author Response

Response to Reviewer 1 Comments 

Point 1: Since it is established in literature that the disruption of TADs and creation of neo-TADs are dominant mechanisms of SVs (e.i. PMID: 25959774; PMID: 27706140) and there are some studies suggesting the involvement of TADs in the limbs development (e.i. PMID: 33497014, PMID: 31511252, PMID: 30475793), it would be interesting to developed a little more the importance of the disruption of TAD as cause for undiagnosed genomic diseases, specially CTEV. 

Response 1: A particularly interesting approach to pursue in future studies will be the study of TAD (Topologically Associated Domain) in patients with CTEV. Recent studies (REF) have shown the involvement of some TADs in limb development, focusing more on gene expression regulation than on coding region mutations. What makes these studies difficult, however, is the fact that although some TADs are identical in each cell line, others are specific to certain tissues and / or certain stages of development. Despite these difficulties, the study of TADs in patients is certainly a way to go to delineate the pathogenesis of CTEV more clearly. 

Point  2: Since you have a well-characterized group of patients, there are several genes already associated to limbs development, and WGS is currently a reliable and relatively robust methodology to be applied in clinical diagnosis, after the laboratory goes through a stage of validation and confirmation of pathogenic variants by other methodologies, it would be interesting to apply it to this group. 

Response 2: Thank you very much. we have added in the conclusions the suggested concept on conclusions (lane 434-437): “To do this and analyze variants outside the coding region of genes as well as structural anomalies, Whole Genome Sequencing (WGS) technology has already proved very useful on a large number of complex pathologies and will certainly be applied, in the future, also in the context clubfoot.” 

Point  3: All the text should be revised, since it were detected some inaccuracies: 

line 55, 83, 96, 151 - small mistakes 

Response 3: 

 -Line 55: I replace “equines” to “equinus” 

-Line 83: I replace “idiopathic congenital talipes equinovarus” with “Idiopathic Congenital Talipes Equinovarus” 

-Line 96: I replace “whose” with “which” 

-Line 151: I replace “statement” with Statement 

-line 149 to 155 – harmonize the way to designate the departments (lower/capital letters): I change lower to capital letter 

-line 163 and 164 - harmonize the way how to refer the tables (Table number or just Table) : I delete “number” 

-line 179 - the table 1 refereed corresponds to “Variants identified in TBX4 gene. All are predicted as benign or likely benign. Three in coding region (Two synonymous and one missense) and one intronic. MAF: Minor Allelic Frequency” not making sense in the text.: This is the caption of Table1.  

-line 265 to 266 - the sentence doesn't make sense: We corrected the sentence 

Point 4: 

Supplementary material

Point 4a: The "Table S2 - Family’s cases" is confused and difficult to consult. Should be improved. 

Response 4a: We improved the Table S2 by inserting a new column indicating the family to which each patients belongs. I have also done for each family the respective pedigree to better understand the kinship of the samples represented in figure S1, always reported in the supplements. I will attach the respective files of both Tables (S2 and S3) and Figure S1. 

Point 4b: The "Table S3 - Isolated probands" should have a logical order for easy consultation (e.i., by order of "cod_pt", for example). 

Response 4b: In table S3 the samples were sorted according to the pt_code as required 

Reviewer 2 Report (New Reviewer)

This work is devoted to the search for mutations in two genes in 162 Italian patients with clubfoot who have a family history of this disease. To date, not even in all family cases of the disease, it is possible to establish a genetic cause. There are some comments to the presentation, so I would recommend shortening the introduction by transferring some of the information presented to the discussion. In addition, it seems to me that a table with unambiguously benign variants in the results is superfluous - just specify them in the text. On the other hand, it would be interesting to look at the number of patients with complex alleles for these genetic variants and the segregation of these alleles in families.

The authors indicate that they did not find any CNVs in the regions of the TBX4 and PITX1 genes. It would be interesting to know about all the CNVs identified in this study, since clubfoot can be a manifestation of a more complex syndrome. In line 134, a typo was made in the number of patients - 161, whereas in all other sections we are talking about 162 patients.

As a result of this work, neither small mutations in the PITX1 gene nor extended deletions and duplications capturing the TBX4 gene were detected in any of the subjects. The negative result of the search for mutations in these genes on a carefully formed sample is worthy of publication.

Author Response

Response to Reviewer 2 Comments

Point 1a: This work is devoted to the search for mutations in two genes in 162 Italian patients with clubfoot who have a family history of this disease. To date, not even in all family cases of the disease, it is possible to establish a genetic cause. There are some comments to the presentation, so I would recommend shortening the introduction by transferring some of the information presented to the discussion.

Response 1a:  The introduction has been shortened by bringing back some of the comments to the discussion as suggested.

Point 1b: In addition, it seems to me that a table with unambiguously benign variants in the results is superfluous - just specify them in the text.

Response 1b: The table 1 with the variants identified on TBX4 gene was deleted from the text and reported in supplementary data as Table 4S, as suggested.

Point 1c: On the other hand, it would be interesting to look at the number of patients with complex alleles for these genetic variants and the segregation of these alleles in families.

Response 1c: Certainly, it is a good suggestion. Although it is not very simple, we are trying to collect the relative’s DNA of our patients to verify the presence and the segregation of complex alleles.

Point 2a: The authors indicate that they did not find any CNVs in the regions of the TBX4 and PITX1 genes. It would be interesting to know about all the CNVs identified in this study, since clubfoot can be a manifestation of a more complex syndrome.

Response 2a: In first analysis, as we explain on text, we focused on the search for CNV on TBX4 and PITX1 genes. Excluding both in our cohort of patients, our future intent is the analysis and the identification of new as a possible cause of clubfoot onset. But, of course, the assessment of structural changes, particularly in areas not covered by genes, their confirmation and the subsequent study of their pathogenicity takes time.

Point 2b:  In line 134, a typo was made in the number of patients - 161, whereas in all other sections we are talking about 162 patients.

Response 2b: I corrected 161 to 162

Point 3: As a result of this work, neither small mutations in the PITX1 gene nor extended deletions and duplications capturing the TBX4 gene were detected in any of the subjects. The negative result of the search for mutations in these genes on a carefully formed sample is worthy of publication.

Response 3: Yes, we agreed. In fact, we made this paper to give this information to the scientific community.

Round 2

Reviewer 2 Report (New Reviewer)

All my comments have been taken into account. I still consider this article worthy of publication, despite the fact that no causes of clubfoot have been identified. The results from a well-formed sample are still very valuable.

Author Response

Thank you for your suggestions in the review process. 

This manuscript is a resubmission of an earlier submission. The following is a list of the peer review reports and author responses from that submission.

Round 1

Reviewer 1 Report

Authors performed an analysis of  PITX1 and TBX4 genes by Sanger sequencing and by SNP-array in patients with CTEV and family members available.

SNP array was used to detect CNV in the studied genes.

Of note, paper is no structured with standard sections and not follow the normal guidelines for a manuscript. Paper writing should be improved with a better genetic vocabulary.

When neither mutations nor CNV were found in 162 patients, analyzing only 2 genes, the next step is to consider performing a WES or, in this case, a GWAS taking advantage of the SNP-Array results. Hence, a genetic association analysis is suggested with SNP-array data or a WES analysis in patients sharing identical phenotype comparing with healthy relatives

Reviewer 2 Report

Review of “What is the exact contribution of PITX1 and TBX4 genes in 2 clubfoot development? A Multicenter Italian study” by Anna M. Bianco

Abstract:

Please correct verbal application in abstract should be past not present….one example ““The aim of this study is to evaluate the prevalence”

Introduction:

Authors mention in introduction” 2–6There are three clinical types of congenital clubfoot: talipes equinovarus, talipes calca neovalgus and metatarsus varus: within them are several subgroups” The bibliographic citation should be done at the end of the sentences. Again in “8 . 8–10In unilateral 67 cases the right foot”

Methodology is insufficiently described, what was described by the authors is not enough for study reproduction or evaluation of design study and statistical analysis methodology…

 Authors did not described how they analysed the pathogenicity of the mutations. Results presentations and absence of clear methodology does not allow for a complete analysis of this manuscript and study relevance….